# Relationship between Cyclic and Non-Cyclic Force-Velocity Characteristics in BMX Cyclists

**DOI:** 10.3390/sports7110232

**Published:** 2019-11-09

**Authors:** Micah Gross, Thomy Gross

**Affiliations:** 1Swiss Federal Institute of Sport, 2532 Magglingen, Switzerland; 2Department of Medicine, Movement and Sport Science, University of Fribourg, 1700 Fribourg, Switzerland; thomy.gross@windowslive.com

**Keywords:** torque, power, sprint cycling, vertical jump, profile

## Abstract

Especially for bicycle motocross (BMX) cyclists, transfer of muscular force-velocity (*Fv*) characteristics between common strength training exercises and cycling is important. This study investigated the relationship between *Fv* characteristics in a common training exercise (squat jumps) and a sport-specific task (cycling) in high-level BMX racers by exploring the degree to which *Fv* and torque–cadence (*Tc*) characteristics correspond. Twelve BMX racers performed an *Fv* (multiple loaded squat jump) and two *Tc* tests (ramp starts and flat-ground sprints). Results revealed very large correlations between $F_{0}$ and $Tor_{0\;start}$ (r = 0.77) and between $P_{max\;jump}$ and $P_{max\;start}$ (r = 0.85). On the other hand, the relationships between $v_{0}$ and $Cad_{0\;start}$ (r = –0.25) and between $S_{Fv}$ and $S_{Tc\;start}$ (r = –0.14) were small and negative. Similar results were observed for sprints. Based on dichotomous classifications (greater or less than group median), several discrepancies occurred, particularly for the profile slopes and high-speed variables. Thus, we recommend performing both jump-based and cycling-specific Fv testing. Of additional note, Tc characteristics on flat ground were similar to, but slightly different from those on the start ramp. Therefore, where possible, *Tc* tests should be carried out on a ramp.

## 1. Introduction

Description of force–velocity (*Fv*) characteristics and the development of methods to monitor and optimize these in the training of competitive athletes have received increasing attention in the sport science literature in recent years [1,2,3,4,5,6,7,8,9]. The physiological mechanism underlying this topic is the inverse relationship between muscle fiber force production and contraction velocity, which was cited in the early 20th century [10]. The application of the *Fv* relationship principle also extends to single and multi-joint contractions, restricted and ballistic movements, as well as simple linear and cyclic actions [6,8,9,11,12,13,14,15]. Even complex locomotive or sport-specific tasks, such as sprint running and cycling, have been shown to display typical *Fv* behavior [8,11,12,16].

Perhaps most commonly, *Fv* characteristics are determined for the lower extremity via vertical jumps with various additional loads, scatter-plotting concentric force versus concentric velocity for each loading condition, and extrapolating a linear model to the force and velocity axis intercepts ($F_{0}$ and $v_{0}$, respectively). Additionally, the slope of the model ($S_{Fv}$) and the theoretical maximal power:(1)$$P_{max}=\frac{F_{0}}{2}\times \frac{v_{0}}{2}$$
are typically obtained [1,4,17]. The same can be done for the acceleration phase of sprint running using models of horizontal force production and sprinting speed at each step [8,16,18] or for sprint cycling using the torque (angular force) and cadence (angular velocity) of each pedal stroke [11,12,19]. Cycling torque–cadence (*Tc*) profiles can be generated with either a series of sprints at different resistances [20] or isokinetic cadences [21], or with a single sprint, where system inertia provides the necessary range of cadences and resistances [19]. The single sprint method, while being efficient and reliable [19], has the additional advantage of being applicable in the field under sport-specific conditions [12,22,23]. As with *Fv* profiles, key parameters defining the *Tc* profile are the two axis intercepts ($Tor_{0}$ and $Cad_{0}$, respectively), the slope ($S_{Tc}$), and the theoretical maximal power ($P_{max}$).

The performance relevant consequence of an athlete’s *Fv* character is the ability to produce power,
(2)P=F×v
at various movement velocities that may be encountered during a particular sport. Depending on the demands of a particular sport, an athlete might seek to optimize his or her *Fv* profile, and thus power–velocity (*Pv*) characteristics, in order to produce the most power at the most relevant velocities, and thereby maximize performance. This is especially important for athletes practicing sports in which performance depends heavily upon maximal power, acceleration, or sprint ability. Moreover, because overloading muscles to a sufficient degree or efficiently completing a certain number of repetitions is sometimes difficult to achieve with sport-specific movements, resistance training exercises are commonly employed as a supplementary means to this end [24,25]. 

Especially now that valid field methods have become available for determining and monitoring individual *Fv* characteristics in common movements—vertical jumping [15] or the bench press [6], for example—interest in individualized training to achieve a certain balance between force and velocity abilities, and thus an optimal *Pv* profile, has grown [2,26]. Explicitly, depending on the individual *Fv* imbalance, training can be focused on the high force or high velocity end of the spectrum [2]. This approach seems to be effective when the task used to determine *Fv* characteristics (e.g., vertical jumping) is also the performance criterion [2]. Sometimes this is not the case, however, and although it seems reasonable that an athlete’s *Fv* character is task-independent, the relationship between *Fv* characteristics between different types of movements performed by the same muscle group has only recently received the attention of researchers [3]. With regards to vertical (jumping) and horizontal (sprinting) *Fv* character, Jimenez-Reyes et al. [3] have shown with a very large sample size that, while these correspond quite well in subjects who are not highly trained, they do not do so in highly-specialized athletes from sports that depend on either jumping or sprinting for performance. In particular, these authors highlight how *Fv* character and performance in sprinters while sprinting are somewhat independent of *Fv* indicators deduced from vertical jumping. Therefore, they discourage making assumptions about one setting based on results from the other, and suggest that athletes be tested in more than one movement. 

An aspect similar to the vertical–horizontal relationship explored by Jimenez-Reyes et al. [3] which is of interest is the relationship between linear movements, commonly used in strength training and cycling. BMX or sprint cyclists, who encounter the *Fv* behavior of cycling each time they accelerate through the range of pedaling cadences, and who also invest a fair amount of time and energy into resistance or weight training, might be especially interested in how *Tc* profiles are related to basic *Fv* characteristics. Such knowledge is the basis for understanding how *Fv* profile adaptations following resistance training are likely to transfer to on-bike torque and power generation across the spectrum of cadences encountered.

Previous investigations have brushed by this topic without thoroughly investigating it. Several studies have shown generally strong correlations between peak mechanical power in a linear task and cycling by significant correlations between jump height or power and peak cycling power [27,28,29,30]. Furthermore, in a series of studies published by Debraux et al. [23,31,32], *Fv* and *Tc* profiles were determined for elite BMX racers. However, linear *Fv* parameters were only compared with peak cycling power or Wingate test performance [32], whereas *Tc* characteristics were compared with 80 m sprint performance only [23,31]. In the study where *Fv* characteristics were reported [32], a *Tc* test was also performed, but correlations between *Fv* and *Tc* parameters (other than $P_{max}$) were not published.

Thus, no previous study has determined by direct comparison whether linear *Fv* characteristics reflect cycling *Tc* characteristics, which is particularly of interest for explosive cycling events, including BMX. Therefore, the aim of this study was to investigate the degree to which *Fv* and *Tc* characteristics are related on an individual basis in BMX racers. The hypothesis was that the relationship between *Fv* characteristics in a linear, technically simple, and familiar training exercise (squat jumps) and a cyclical, technically complex, and sport-specific task (ramp starts and cycling sprints ) is such that riders display similar characteristics (steep or flat profiles, high or low maximal power) in both *Fv* and *Tc* profiles. Additional aims of the study were to compare *Tc* characteristics between ramp starts and flat-ground sprints and to report measures of intra-trial reliability for *Tc* parameters.

## 2. Materials and Methods

### 2.1. Study Participants

Nine male and three female BMX racers from the Swiss national team selection pool participated in the study. Participants (and for those under 18, their parents) received information in advance regarding the aims and procedures of the study and gave their written consent to participate. All study procedures were approved by the ethical review board of the Swiss Federal Institute of Sport (020-LSP) and conformed to international ethical standards. All participants competed at the national or international level in either the junior or elite category at the time of the study. They were injury-free and accustomed to the testing procedures. Their mean (±standard deviation) age, height, and body mass were 24.6 ± 2.6 years, 174.4 ± 8.0 cm, and 72.2 ± 10.5 kg, respectively. 

### 2.2. Design

The study adopted a cross-sectional, descriptive design. For each participant, three tests were conducted on the same day within a window of 2.5 h. Following anthropometric measurements, participants performed, in this order, a vertical jump test, BMX ramp starts, and flat-ground sprints. 

#### 2.2.1. Force–Velocity (Fv) Jump Test

Following an individual warm-up, subjects performed the force–velocity (*Fv*) test, which comprised vertical squat jumps under various loading conditions. Each trial was performed from a static starting position (without countermovement) at a knee angle of 90°. The starting position was determined before the first trial with a goniometer and a horizontal bar positioned under the participant’s glutes, such that they could feel the correct depth for all subsequent trials. Loading conditions ranged from 0–80% of body mass at 20% increments. Loads of 20–80% were provided using a barbell placed across the shoulders. For the unloaded condition, a wooden stick with a mass of 300 g was used in place of the barbell. The order of the loads was randomized and three to five correct trials were performed under each loading condition. Thirty seconds separated trials and three minutes separated loading conditions. Participants were instructed to jump as high and as explosively as possible.

For all jumps, ground reaction force was measured at 1000 Hz with a one-dimensional force plate (MLD Test EVO 2, SP Sport, Trins, Austria). Using the total mass, determined by the force plate prior to each jump, the accompanying software (Muskelleistungsdiagnose 2010, version 5.2.0.6101, InfPro IT Solutions GmbH, Innsbruck, Austria) calculated acceleration–time, velocity–time, power–time, and position–time curves of the center of mass from the recorded force–time signal for each jump. If a negative acceleration (countermovement) was detected within 200 ms of the onset of positive acceleration, the trial was deemed invalid and deleted. For each valid jump, the mean ground reaction force, mean velocity, and mean mechanical power for the concentric phase were retained for further analysis. 

#### 2.2.2. Torque–Cadence (Tc) Cycling Tests 

After the jump test, participants changed clothes and proceeded to the BMX track, where they completed a 15-minute technical warm-up to prepare for the start ramp measurements, the first of two torque–cadence (*Tc*) tests. Riders performed five starts on a supercross ramp (mean slope 26°) with the same electronic, standardized start command, and randomized gate used in international competition, with no other riders on the ramp. Riders were instructed to start as fast as possible until the first jump, which was about 6 m beyond the bottom of the ramp. Starts times were measured from the gate to the end of the ramp (22.7 m) using electronic timing gates (TC Timing System, Brower, Salt Lake City, UT, USA) and mean starting velocity was calculated from time and distance. Start trials were separated by five minutes.

Following the last start, participants had 10–15 min recovery before beginning the flat-ground sprints, the second *Tc* test. Each rider performed three maximal sprints on a level asphalt surface, for which they were instructed to accelerate maximally up to maximum speed (~50 m). Riders started with cranks approximately horizontal and their rear pedal placed on a block to keep their balance, and began the sprint on their own initiative. Sprint trials were separated by five min of recovery.

Riders performed the starts and sprints using their own racing bikes with their usual competition gear ratio. Before the tests, each bike was equipped with a modified crank-based powermeter. The powermeter (Shimano DXR with SRM spider, SRM, Jülich, Germany) was modified with a gyroscope (Axiamo GmbH, Biel, Switzerland), which measured crank angular velocity, intercepted the analogue torque signal from the SRM, and recorded the synchronized data streams at 100 Hz [33]. Crank angular velocity was converted to an equivalent cadence and retained along with torque for further processing. 

### 2.3. Data Processing 

Data processing was performed using customized Python code, including the sklearn.linear_model and scipy.stats modules. For each participant and loading condition, the median jump trial based on average concentric power was selected and its average concentric force (*F*) and velocity (*v*) values were plotted along with those from the other loading conditions in a *v*-*F* scatter plot. Thereafter, a linear regression model was fitted to describe the *Fv* profile. From this model, the characteristic variables ($F_{0}$, $v_{0}$, $S_{Fv}$, and $P_{max}$) were retained for further analysis.

For each start and sprint trial, means of torque (*Tor*) and cadence (*Cad*) were calculated for each complete pedal downstroke, i.e., each half revolution of the cranks (the first quarter revolution, because cranks are initially horizontal, was not comparable to other strokes and was therefore excluded). Pedal downstrokes were identified by nadirs in the torque signal, which corresponded approximately to the vertical crank arm positions. Mean *Tor* and *Cad* for each complete pedal downstroke in the phase of continually increasing cadence were then plotted in a *Cad*-*Tor* scatter plot and a linear regression model was fitted to describe the *Tc* profile. From the two trials with the highest r^2^ values, the characteristic variables ($Tor_{0}$, $Cad_{0}$, $S_{Tc}$, and $P_{max}$) were extracted and averaged for further analysis.

In addition to the key variables from Fv and Tc profiles, additional parameters were calculated to describe the nature of linear modeling in each test setting. These included the measurement range of F, v, Tor, and Cad, the extrapolation range from the nearest measurement to its respective axis intercept, and the ratio between the two (extrapolation ratio, i.e., of the extrapolation range to the measurement range).

### 2.4. Statistical Analyses

Normal distribution of all variables was assessed with scipy.stats.normaltest and a *p*-threshold of 0.05; this test is based on D’Agostino and Pearson’s test that combines skew and kurtosis to produce an omnibus test of normality. The typical error of each *Tc* profile’s variables was assessed across the five start trials or three sprint trials using customized spreadsheets [34]. For the jumps, raw data points of the individual jumps were used to assess the typical errors of concentric *F* and *v* at the five different loading conditions.

To evaluate the linear association between pairs of normally distributed variables, Pearson’s correlation coefficients (r) were calculated and expressed along with 95% confidence limits (c.l.). In the case of a non-normally distributed variable, a correlation between the intra-group ranks of both variables was performed. Based on these the absolute values of r |r|, correlations were interpreted qualitatively as either trivial (|r| < 0.1), small (|r| = 0.1–0.3), moderate (|r| = 0.3–0.5), large (|r| = 0.5–0.7), very large (|r| = 0.7–0.9), or extremely large (|r| > 0.9) [35]. Additionally, participants were classified for each *Fv* and *Tc* variable as lying above or below the group median, and the fraction of participants having divergent classifications for a *Tc* variable and the analogous *Fv* variable was reported.

Also, using both Pearson’s correlation analysis and repeated-measures t-testing, *Tc* variables were compared between the ramp starts and flat-ground sprints, to see if and how test setting affected the *Tc* profile. Finally, correlations between $P_{max}$ in each of the three test settings and their respective contributing parameters $F_{0}$/$Tor_{0}$ and $v_{0}$/$Cd_{0}$ were explored.

## 3. Results

All 12 participants completed the jump test and the flat-ground sprints. However, one female participant was unable to perform the ramp starts and one subject’s sprint data were missing due to a malfunction of the crank. Therefore, the number of data points for these measurements was reduced to 11.

Data describing the construction of *Fv* and *Tc* profiles are displayed in Table 1 and Table 2, while example profiles are portrayed in Figure 1.

Typical errors of raw concentric *F* and *v* during jumps under various loading conditions ranged from 28 to 40 N (1.4–2.6%) and 0.02 to 0.04 m·s^−1^ (2.3–5.1%), respectively. The typical errors of *Tc* characteristics are displayed along with the descriptive data (Table 3).

The participants’ *Fv* characteristics and *Tc* characteristics are displayed in Table 3. Only $S_{Fv}$ from the vertical jump test did not pass the test for normal distribution. Therefore, correlations with this variable were done with intra-group ranks, whereas all other correlations were performed with actual values.

Results (Figure 2) revealed very large correlations between $F_{0}$ and $Tor_{0\;start}$ and between $P_{max\;jump}$ and $P_{max\;start}$. On the other hand, relationships between $v_{0}$ and $Cad_{0\;start}$ and between $S_{Fv}$ and $S_{Tc\;start}$ were both small and negative. A somewhat similar situation was observed when comparing *Fv* variables with the *Tc* variables obtained from the flat-ground sprints—whereas very strong relationships existed between $F_{0}$ and $Tor_{0\;sprint}$ and between $P_{max\;jump}$ and $P_{max\;sprint}$, the relationship between $v_{0}$ and $Cad_{0\;sprint}$ was trivial and the correlation between $S_{Fv}$ and $S_{Tc\;sprint}$ was small to moderate.

When participants were classified as lying above or below the group median for each of the *Fv* and *Tc* parameters from the ramp starts (Figure 3), 4 out of 11 riders had divergent classifications for $Tor_{0\;start}$ and $F_{0}$, although 3 of these lay within 1% of the median for one of the parameters. When comparing $P_{max\;start}$ with $P_{\max jump}$, no riders had divergent classifications. On the other hand, 7 out of 11 riders had divergent classifications for $Cad_{0\;start}$ and $v_{0}$, as did 6 riders for $S_{Tc\;start}$ and $S_{Fv}$. When classifications were based on *Tc* characteristics from flat-ground sprinting instead of ramp starts, 2 out of 11 riders had divergent classifications for each of $F_{0}$ versus $Tor_{0\;sprint}$ and $P_{\max jump}$ versus $P_{\max sprint}$. For each of $S_{Fv}$ versus $S_{Tc\;sprint}$ and $v_{0}$ versus $Cad_{0\;sprint}$, 4 out of 11 riders had divergent classifications.

A comparison of the two *Tc* test settings revealed large to extremely large correlations between Tc properties (r = 0.86, 0.57, 0.51, and 0.97 for $Tor_{0}$, $Cad_{0}$, $S_{Tc}$, and $P_{max}$, respectively; *p* < 0.05 for $Tor_{0}$ and $P_{max}$). Furthermore, no significant differences between the two settings were found for $Tor_{0}$, $Cad_{0}$, or $S_{Tc}$ (*p*-values 0.31–0.79). On the other hand, as a product of $Tor_{0}$ and $Cad_{0}$ (for both of which the group mean was slightly higher during start ramp trials) $P_{max}$ was significantly greater (*p* = 0.01) in start ramp trials than flat-ground sprints. Whereas $F_{0}$ and $v_{0}$ correlated equally with $P_{\max\;jump}$ (r = 0.73 and 0.72, respectively), and $Tor_{0\;sprint}$ and $Cad_{0\;sprint}$ correlated similarly with $P_{\max\;sprint}$ (r = 0.99 and 0.89, respectively), $Tor_{0\;start}$ tended (*p* = 0.06) to correlated more strongly (r = 0.91, 95% c.l.: 0.68–0.98) with $P_{\max\;start}$ than did $Cad_{0\;start}$ (r = 0.52, 95% c.l.: −0.12–0.85).

## 4. Discussion

To the best of our knowledge, this was the first study to investigate whether accustomed and well-trained BMX cyclists have similar force–velocity characteristics for both a linear, technically simple exercise and for a cyclic, technically complex but sport-specific task. The hypothesis was tested using loaded squat jumps, ramp starts, and flat-ground sprints on subjects’ BMX bikes. From the perspective of this group of athletes, understanding how linear *Fv* and *Pv* characteristics relate to on-bike *Tc* and *Pc* behavior is essential in order to get the greatest possible benefit from resistance training aimed at optimizing muscle power. Although the current participants would be primarily interested in power output during a ramp start, a flat-ground sprint was included for the sake of comparison. 

Based on Pearson’s correlations, there were clear relationships between theoretical maximal force ($F_{0}$) and theoretical maximal torque ($Tor_{0\;start}$ or $Tor_{0\;sprint}$), as well as between $P_{\max jump}$ and both $P_{\max start}$ and $P_{\max sprint}$. This being the case, one might expect the other key profile characteristics to correlate as well with one another. However, correlations between profile slopes ($S_{Fv}$ and $S_{Tc}$) and between $v_{0}$ and $Cad_{0}$ were not nearly as strong, and were even negative in the case of ramp starts. 

Based on less stringent, dichotomous classifications (greater or less than the group median), generally good agreement was observed for the parameters $F_{0}$/$Tor_{0}$ and $P_{\max}$. On the other hand, we found poor agreement between classifications based on $v_{0}$/$Cad_{0}$ and profile slopes. The largest discrepancies in these cases appeared for the riders labeled 1, 7, and 8, found in the upper left quadrants of Figure 3b,c. Although these riders had comparatively lower $v_{0}$ and steeper Fv profiles, they displayed flatter Tc profiles and higher $Cad_{0}$ during ramp starts than most of their counterparts. Thus, while a dichotomous classification of *Fv* parameters could indicate *Tc* character in some cases, it must be acknowledged that a large percentage of riders in the present study posed exceptions to the general rule. Consequently, there is some risk of falsely diagnosing individuals’ *Tc* character, particularly on the speed end of the profile, when drawing upon *Fv* data. 

Thus, the present study confirmed previous findings [27,28,29,30,32] that peak jumping power ($P_{\max jump}$) is strongly correlated with cycling $P_{max}$ taken from either a flag-ground sprint test or a ramp start. Additionally, the present study showed for the first time that BMX riders with a high $F_{0}$ in their vertical jump-based *Fv* profile are most likely to have a high $Tor_{0}$ in their *Tc* profile and thus be able to produce high torque at the lowest cadences, such as at the onset of a ramp start. Incidentally, power in the first pedal stroke, at the lowest cadence, is one of the most decisive mechanical parameters for BMX starting performance [33]. This would suggest that high maximal strength at low movement velocities, as indicated by $F_{0}$ and $Tor_{0}$, is rather important for BMX start performance. Although the $v_{0}$ from vertical jump-based *Fv* profiles did not line up consistently with $Cad_{0}$, this parameter was shown to be less decisive for BMX starting performance than $Tor_{0}$ [33].

The lack of a positive relationship between $v_{0}$ and $Cad_{0\;start}$ and between $S_{Fv}$ and $S_{Tc\;start}$ could be explained by divergent technical demands of ramp starts and vertical jumps. Namely, the attainment of a high peak cadence during a ramp start might be more limited by technical skill than by basic neuromuscular ability. This supposition would be in line with the reasoning of Jimenez-Reyes et al. [3], who offered the technical demands of sprinting as an explanation for poor agreement between vertical jumping and horizontal sprinting Fv profiles in high-level sprinters. In the present study, the fact that there was a small to moderate correlation between profile slopes when jumps were compared to technically less-demanding flat-ground sprints offers further evidence that technical, rather than neuromuscular ability, is likely a limiting factor at high speeds during ramp starts. 

Thus, given the present findings, $P_{\max}$ obtained from a vertical-jump based Fv profile could be a good indicator of the general performance level of BMX racers. Furthermore, $P_{\max}$ obtained from a ramp-start based Tc profile tends to vary more due to differences in $Tor_{0}$ (extremely strong correlation) than in $Cad_{0}$ (moderate correlation), which suggests that, in practice, $Tor_{0}$ is also a good indicator of general performance level. On the other hand, individual strengths and weaknesses with regard to optimal Fv profiles and Fv imbalances [2,26,30] must take the high-velocity end of the spectrum into account as well. In the case of BMX ramp starts, the high-velocity end of the spectrum is not well described by $v_{0}$ obtained from vertical jumps. Therefore, and because optimal Tc characteristics for BMX start performance are probably individual [36], BMX racers should obtain Tc parameters on the start ramp if possible. In this manner, the technical or movement-specific contributors to the Tc profile are better accounted for. Nonetheless, jump-based Fv parameters obtained in combination with start-specific Tc characteristics could be useful for recognizing neuromuscular potential, which may be limited by technical or movement-specific deficits. Such might be the case for a rider with a comparably high $v_{0}$ but a low $Cad_{0}$ while starting, whereas a rider in the opposite situation might be limited by basic neuromuscular ability. The next best option to ramp starts could be a flat-ground sprint, since moderate to very strong correlations between these Tc parameters and those from ramp starts were found. However, caution should be taken when making assumptions about Tc character based on vertical-jump based Fv testing alone. 

One factor which may have limited our ability to recognize positive relationships between cyclic and acyclic abilities is a methodological shortcoming of this study, whereby the data points used for generating *Fv* profiles fell within a rather small range compared to the range over which the linear regression line had to be extrapolated to obtain $F_{0}$ and $v_{0}$ (Figure 1a, Table 2). This was necessary for practical purposes, as vertical jumps cannot be easily performed with lighter or much heavier loads than those we employed. Although similar loaded jump protocols have been employed in other studies [1,4,17], this shortcoming—namely, the robustness of an extrapolated point on a linear regression line that lies three to four times as far away from the measured data points as the range of these data points is wide—is seldom discussed. We attempted to make *Fv* linear models as robust as possible by taking multiple measurements at each loading condition and selecting only the most representative trial (that corresponding to the median concentric power) at each loading condition. The typical errors of the *Fv* model inputs *F* (1.4–2.6%) and *v* (2.3–5.1%) themselves were rather small, but could have nonetheless produced inaccurate *Fv* characteristics because of the large degree of extrapolation. *Tc* profiles, on the other hand, were based on data points spanning a much larger range relative to the extrapolated axis intercepts (Figure 1b and 1c, Table 2), which presumably allowed for more robust profiles, as is reflected in the small typical errors (3–10%) for *Tc* end parameters ($Tor_{0}$, $Cad_{0}$, $S_{Tc}$, $P_{\max}$). A further limitation of the current study, which may have influenced results, is the somewhat small sample size. Therefore, the overall generalizability of our findings is not known.

## 5. Conclusions

In conclusion, force–velocity and torque–cadence characteristics derived from vertical jumps and cycling accelerations appear to be somewhat related in athletes who are well accustomed to both tasks. In particular, the ability to produce force at low velocities and the modeled maximal power correspond quite consistently between tasks. However, parameters at the high-velocity end of profiles did not agree consistently, which, importantly, precludes drawing definitive conclusions about individuals’ *Tc* character based on *Fv* profiles. Thus, although resistance training or other interventions with the potential to shift *Fv* characteristics remain useful for optimizing performance in BMX (as well as other explosive cycling disciplines), technical or movement-specific aspects of BMX starts must be acknowledged as well while pursuing an optimal Tc balance. 

## Figures and Tables

**Figure 1 sports-07-00232-f001:**
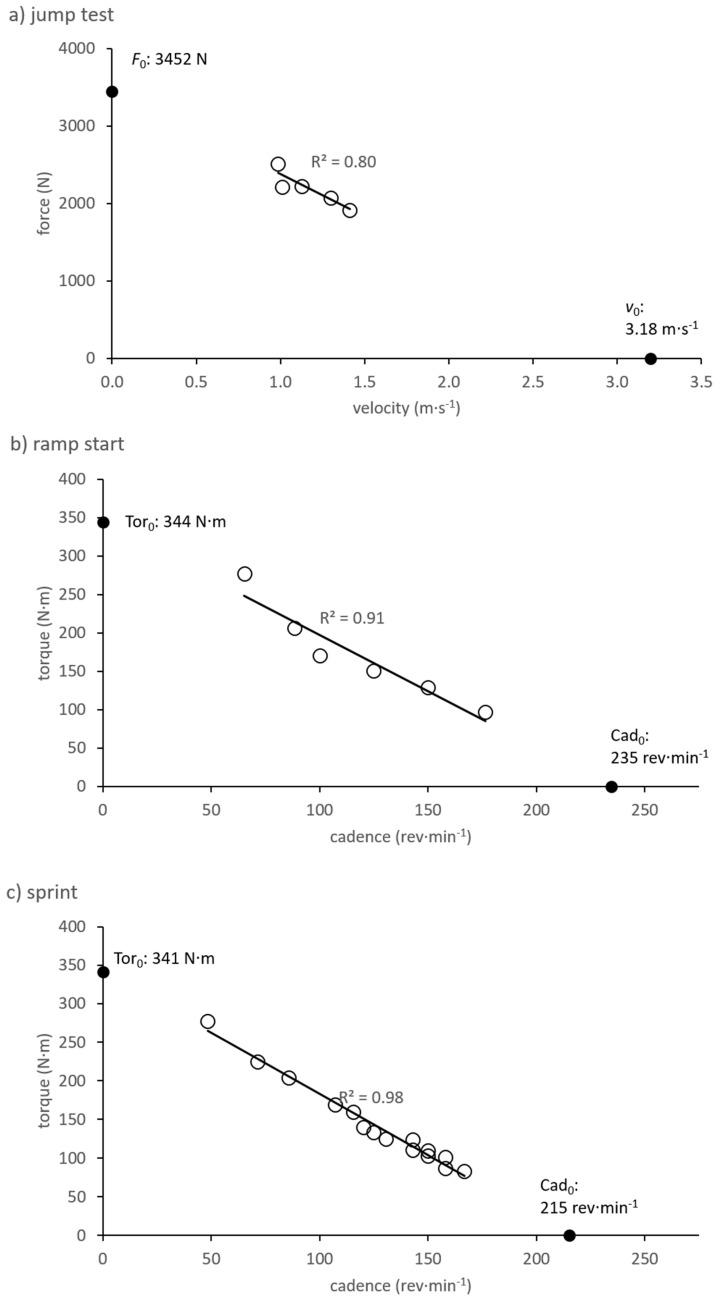
Example profiles from one participant. (**a**) force–velocity (*Fv*) profile from the vertical jump test, (**b**) torque–cadence (*Tc*) profile from a ramp start, (**c**) *Tc* profile from a flat-ground sprint. Filled points indicate the axis intercept values of linear regression models.

**Figure 2 sports-07-00232-f002:**
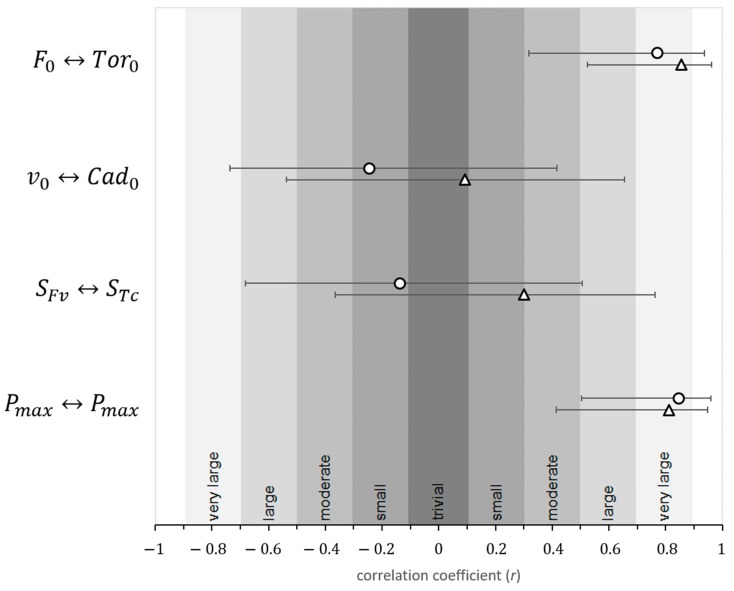
Correlation coefficients between force-velocity characteristics ($F_{0}$, $v_{0}$
*,*$\;S_{Fv}$
*,* and $P_{max}$ ) from a vertical jumping test and torque–cadence characteristics ($Tor_{0}$, $Cad_{0}$, $S_{Tc}$, and $P_{max}$ ) from BMX ramp starts (circles) and flat-ground sprints (triangles) on a BMX bike. Error bars represent 95% confidence intervals of the correlation coefficients.

**Figure 3 sports-07-00232-f003:**
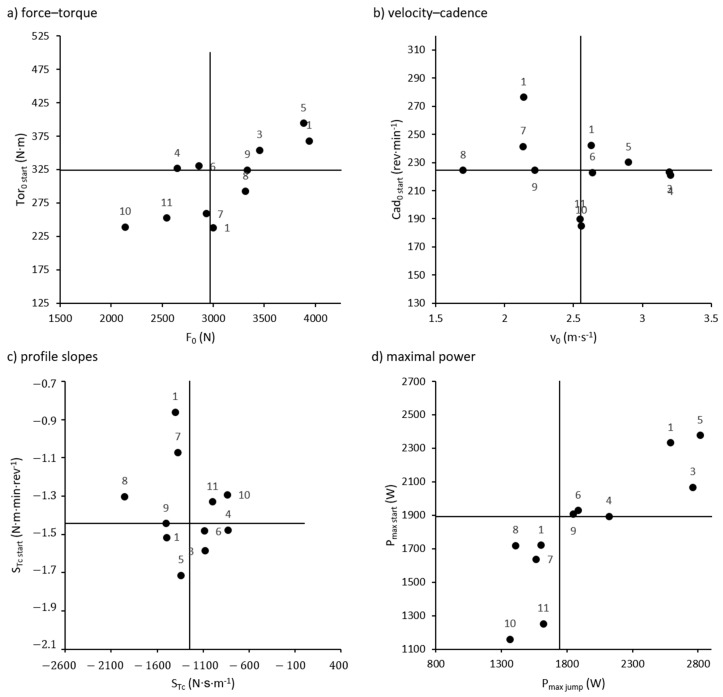
Comparison of dichotomous classifications for force–velocity (*Fv*) characteristics from the vertical jumping test and torque–cadence (*Tc*) characteristics from the ramp starts, placing each participant (points) in one of four quadrants. Right side quadrants of ***a***–***d***: *Fv* characteristic greater than the group median. Upper two quadrants of ***a*–*d***: *Tc* characteristic greater than group median. Numbers on data points represent participants’ intra-group rankings in terms of ramp start performance (mean velocity).

**Table 1 sports-07-00232-t001:** Parameters describing the construction of force–velocity (*Fv*) and torque–cadence (*Tc*) profiles.

Descriptive Parameter	Value	*Fv*	*Tc* Start	*Tc* Sprint
number of data points (loading conditions or pedal strokes)	mean ± s.d.	4.8 ± 0.6	6.3 ± 1.0	15.8 ± 3.0
	minimum	3	4	10
	maximum	5	7	20
*r*^2^ of regression	mean ± s.d.	0.92 ± 0.08	0.91 ± 0.04	0.95 ± 0.03
	minimum	0.72	0.87	0.86
	maximum	1.00	0.97	0.98

*Fv* refers to the force–velocity model of vertical squat jumps under different loading conditions. *Tc* start and *Tc* sprint refer to torque–cadence models of maximal cycling accelerations on the BMX starting ramp and on flat ground, respectively. s.d.: standard deviation.

**Table 2 sports-07-00232-t002:** Measurement and extrapolation ranges of the force–velocity (*Fv*) and torque–cadence (*Tc*) profiles

Variable	Value	Unit	*Fv*	*Tc* Start	*Tc* Sprint
*v* or *Cad*	minimal measured value	(% *v*_0_ or *Cad_0_*)	35 ± 5	29 ± 2	24 ± 1
	maximal measured value	(% *v*_0_ or *Cad_0_*)	49 ± 5	73 ± 5	78 ± 4
	measurement range width	(m·s^−1^ or rev·min^−1^)	0.37 ± 0.07	99 ± 13	118 ± 16
	extrapolation range width	(m·s^−1^ or rev·min^−1^)	1.28 ± 0.35	60 ± 18	47 ± 11
	extrapolation ratio	(extrap. range width: meas. range width)	3.4 ± 0.5	0.6 ± 0.2	0.4 ± 0.1
*F* or *Tor*	minimal measured value	(% *F*_0_ or *T_0_*)	49 ± 5	28 ± 3	24 ± 2
	maximal measured value	(% *F*_0_ or *T_0_*)	65 ± 6	77 ± 3	85 ± 3
	measurement range width	(N or N·m)	466 ± 74	152 ± 41	176 ± 33
	extrapolation range width	(N or N·m)	1081 ± 311	69 ± 9	42 ± 14
	extrapolation ratio	(extrap. range width: meas. range width)	2.3 ± 0.6	0.5 ± 0.1	0.2 ± 0.1

Data are presented as mean ± standard deviation. *Fv* refers to the force–velocity model of vertical squat jumps under different loading conditions. *Tc* start and *Tc* sprint refer to torque–cadence models of maximal cycling accelerations on the BMX starting ramp and on flat ground, respectively. *F*_0_, *v*_0_: force (*F*) and velocity (*v*) axis intercepts, respectively, of the linear force–velocity (*Fv)* model. *Tor*_0_, *Cad*_0_: torque (*Tor*) and cadence (*Cad*) axis intercepts and slope, respectively, of the linear torque–cadence (*Tc*) model.

**Table 3 sports-07-00232-t003:** Summary of force–velocity and torque–cadence characteristics.

Parameter	Mean ± s.d.	Minimum	Maximum	Typical Error	Percentage of Typical Error (%)
jump test, *n* = 12					
*F*_0_ (N)	3067 ± 538	2137	3938	-	-
*v*_0_ (m·s^−1^)	2.5 ± 0.4	1.7	3.2	-	-
*S_Fv_* (N·s·m^−1^)	−1255 ± 323	−1953	−826	-	-
*P*_max jump_ (W)	1935 ± 519	1365	2815	-	-
ramp start, *n* = 11					
*Tor**_0 start_* (N·m)	307 ± 54	238	394	21	7.1%
*Cad**_0 start_* (rev·min^−1^)	226 ± 25	185	276	19	8.6%
*S_Tc start_* (N·m·min·rev^−1^)	−1.37 ± 0.24	−1.71	−0.86	0.16	12%
*P*_max start_ (W)	1817 ± 383	1159	2376	67	3.7%
sprint *n* = 11					
*Tor*_0 sprint_ (N·m)	290 ± 51	221	365	14	4.8%
*Cad*_0 sprint_ (rev·min^−1^)	218 ± 12	199	239	11	5.1%
*S_Tc sprint_* (N·m·min·rev^−1^)	−1.32 ± 0.18	−1.55	−1.09	0.11	8.2%
*P**_max sprint_* (W)	1662 ± 365	1149	2285	43	2.6%

s.d.: standard deviation. *F*_0_, *v*_0_, S*_Fv_*: force (*F*) and velocity (*v*) axis intercepts and slope, respectively, of the linear force–velocity (*Fv*) models based on concentric *F* and *v* during vertical squat jumps under various loading conditions. *P*_max jump_: theoretical maximal jumping power (defined as ½F_0_ × ½v_0_). *Tor*_0_, *Cad*_0_, *S_Tc_*: torque (*Tor*) and cadence (*Cad*) axis intercepts and slope, respectively, of the linear torque–cadence (*Tc*) models based pedal strokes during ramp starts and flat-ground sprints. *P*_max start_, *P*_max sprint_: theoretical maximal cycling power (defined as power at ½*Tor*_0_ and ½*Cad*_0_).

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
