# Peer review of "Relationship between Cyclic and Non-Cyclic Force-Velocity Characteristics in BMX Cyclists"

_sports, 2019, doi:10.3390/sports7110232_

Round 1

Reviewer 1 Report

This manuscript is well-written; however, additional changes are needed.  The content overall will contribute well to the existing literature regarding the force-velocity relationship; however, the authors should expand the practical applications of the study.  Please consider the below comments.

Line 30: Omit 'already'

Line 50: Using 'his or her' twice in the same sentence is redundant.  Modify this sentence.

Line 66: Consider adding a brief discussion of Jiménez-Reyes et al. Relationship between vertical and horizontal force-velocity-power profiles in various sports and levels of practice. PeerJ 2018

Lines 68-71: Sentence fragment.  Modify this sentence.

Lines 80-81: Which study are the authors referring to here?

Line 108: Which posterior are the authors referring to?  Glutes?  Hamstrings?

Line 134: What type of surface was used for the sprints? This is important to note so that other researchers could replicate this study.

Line 150: What variables were used? Peak force? Force at peak power? Peak velocity? Velocity at peak power? This is currently unclear.

Lines 280-281: Wouldn't this be dependent on their individual relative strength levels?

The authors should add a limitations section to outline any potential shortcomings of the current study.  For example, a small sample size was used which may influence the magnitude of the relationships in the current study.

In addition, the authors should outline how the information of the current study can be used by practitioners and researchers to develop future research projects and improve their athletes' performance?  Further practical application is needed.

Reviewer 2 Report

GENERAL COMMENTS

This study explored the relationship between the FV parameters of a squat jump exercise, and the Torque-Cadence (Tc) relationship of two cycling exercise. The approach is interesting but I have two main concerns:

Statistical analysis is based on the called “Magnitude Based Inference” technique. In this regards a lot of criticism about this technique have been recently published and its use has not been recommended (see for example PMID: 31149752; PMID:29683920) The method for obtaining FV and TC, removing the data that reduced the goodness of fit of the linear model is not sound. In my opinion it could have biased the analysis, removing curvilinear tendencies that could have been observed in some subjects.

Specific comments: Lines Comment

L 22. This final conclusion does not seem to be based on the results reported in the abstract. Please, qualify it

L 57-88. All these paragraphs refer to transfer but in fact you only checked if the parameters obtained from a linear movement were in some way related with those obtained from a cycling test, but not about how the improvement of FV of a linear movement was "transferred" to a cycling performance. I recommend rewriting this part of the introduction.

L 147-150. Which value did you select: peak, mean force…?

L 151-166. As previously said in the general comments, the procedure for obtaining the linear relationships seem to be designed for forcing the FV and TV relationships to be linear. Several studies have shown that some subjects present a curvilinear profile, so deleting points (when the movement is correctly performed) on the basis of being away from the linear tendency, could, in my opinion, have biased the results. I would recommend reporting the R2 values (median and range) before after removing “outliers” records. Furthermore, could you report how many points did you remove? On the other hand, what was the rationale for selecting a R2 of 0.9 and not for example 0.95 or 0.85? Please, clarify it

L170-172. Typical error is used to evaluate reliability but this was not one of the objectives of this study.

L 175. Do you mean: to evaluate the "linear association"...??

L 180-184. Please, see my general comment regarding MBI

L 197. Table 2. Al these variables should be explained in the text. On the other hand I think that the 6th and 7th row refer to % of F0 or T0 but not to V0 or C0. Please, amend it.

L 197-200. As previously said, I don´t really understand why you report the typical error. Is it related with any of the objectives of the study?

L 208-215. Please report p-values and 95% CI

L 216. Figure 2. How did you select the threshold for small, trivial...values?

L 281-284. You should refer in the discussion how removing some points for obtaining high R2 could have affected to your results.

Round 2

Reviewer 2 Report

The authors must be commended for the nice job performed in order to improve the manuscript. All of my suggestions were  correctly addressed